# Machine Learning Applied to a Dual-Polarized Sentinel-1 Image for Wind Retrieval of Tropical Cyclones

Yuyi Hu [1], Weizeng Shao [1,*], Wei Shen [2], Yuhang Zhou [1] and Xingwei Jiang [3]

1. College of Marine Sciences, Shanghai Ocean University, Shanghai 201306, China; d210200046@st.shou.edu.cn (Y.H.); shenwei@wti.ac.cn (Y.Z.)
2. China Waterborne Transport Research Institute, Ministry of Transport, Beijing 100088, China; m220200672@st.shou.edu.cn
3. National Satellite Ocean Application Service, Ministry of Natural Resources, Beijing 100081, China; xwjiang@mail.nsoas.org.cn
* Correspondence: wzshao@shou.edu.cn; Tel.: +86-21-61900326

**Abstract:** In this work, three types of machine learning algorithms are applied for synthetic aperture radar (SAR) wind retrieval in tropical cyclones (TCs), and the optimal method is confirmed. In total, 30 Sentinel-1 (S-1) images in dual-polarization (vertical–vertical [VV] and vertical–horizontal [VH] were collected during the period from 2016 to 2021, which were acquired in interferometric-wide and extra-wide modes with pixels of 10 m and 40 m, respectively. More than 100,000 sub-scenes with a spatial coverage of 3 km are extracted from these images. The dependences of variables estimated from sub-scenes, i.e., VV-polarized and VH-polarized normalized radar cross-section (NRCS), as well as the azimuthal wave cutoff wavelength, on wind speeds from the stepped-frequency microwave radiometer (SFMR) and the soil moisture active passive (SMAP) radiometer are studied, showing the linear relations between wind speed and these three parameters; however, the saturation of VV-polarized NRCS and the azimuthal wave cutoff wavelength is observed. This is the foundation of selecting input variables in machine learning algorithms. Two-thirds of the collocated dataset (20 images) are used for training the process using three machine learning algorithms, i.e., eXtreme Gradient Boosting (XGBoost), Multi-layer Perceptron, and K-Nearest Neighbor, and the coefficients are fitted after training completion through 20 images collocated with SFMR and SMAP data. Another 10 images are taken for validation up to 70 m/s, yielding a 2.53 m/s root mean square error (RMSE) with a 0.96 correlation and 0.12 scatter index (SI) using XGBoost. The result is better than the >5 m/s error achieved using the existing cross-polarized geophysical model function and the other two machine learning algorithms; moreover, the comparison between wind retrievals using XGBoost and Level-2 CyclObs products shows about 4 m/s RMSE and 0.18 SI. This suggests that the machine learning algorithm XGBoost is an effective method for inverting the TC wind field utilizing SAR measurements in dual-polarization.

**Keywords:** tropical cyclone; wind; synthetic aperture radar

## 1. Introduction

Tropical cyclones (TCs) associated with extreme waves and heavy rainfall are huge ocean disasters that cannot be ignored in coastal waters; moreover, TCs are an important way of heat exchange in mid-to-high latitude regions [1]. The on-scene observations in TCs are difficult due to the danger and extreme sea states. Instead, numeric modeling [2] is popularly employed for the hindcasting research on TCs, i.e., their structure, track, intensity, and strong wind-generated waves [3,4]. In addition, spaceborne satellites carrying optical and microwave sensors can be used for TC monitoring in near real-time, i.e., scatterometers [5], polarimetric microwave radiometers [6], and altimeters [7]; however, these remote-sensed products do not meet the requirements of research on the atmospheric and marine dynamics at the submeso- and meso-scales.

Synthetic aperture radar (SAR) has the capability of sea surface dynamics monitoring and target detection in all weather conditions, i.e., winds [8,9], waves [10,11], oceanic fronts [12], oil spills [13] and ships [14]. Although basic TC parameters, i.e., the geographic location of its eye, morphology [15], and intensity [16], are easily obtained on co-polarized (vertical–vertical [VV] and horizontal–horizontal [HH]) SAR, the detailed structure of a wind profile is difficult to retrieve from co-polarized SAR due to the saturation problem in strong winds [17,18]. It is revealed that the SAR backscattering signal in cross-polarization (vertical–horizontal [VH] and horizontal–vertical [HV]) does not easily encounter saturation with increasing wind speed [19,20]; therefore, TC wind information is usually retrieved from the cross-polarized SAR image. Optionally, the TC wind field is also retrieved using the SAR-measured wave cutoff wavelength in the azimuth/flight direction [21,22] or the SAR-derived wave parameters [23].

A geophysical model function (GMF) was developed for SAR wind retrieval [24,25], which relates the normalized radar cross section (NRCS) with a wind vector and incidence angle. As for cross-polarized SAR, it is found that the dependence of the wind direction on NRCS is quite weak; therefore, the cross-polarized GMF is simplified to the function of wind speed at a fixed incidence angle [26]. Several GMFs in cross-polarization are adopted for C-band SARs, i.e., RADARSAT-2 (R-2) [27], Sentinel-1 (S-1) [28], and Gaofen-3 (GF-3) [29]. These GMFs are able to be applied for SAR wind retrieval under extreme weather conditions up to 70 m/s; however, the accuracy (i.e., ~5 m/s error) [30] is worse than the 2 m/s root mean square error (RMSE) of wind speed achieved using the co-polarized GMF (i.e., latest version CMOD7 of the CMOD family) [31] under regular atmospheric conditions [32]. In order to enlarge the spatial coverage, SAR typically uses a wide scanning mode to monitor a TC, i.e., interferometric-wide (IW) and extra-wide (EW) for S-1. Specifically, several sub-swaths are paired into an IW and EW image. Due to different noise-equivalent sigma zero (NESZ), the discontinuity of the wind retrieval at the edge of sub-swaths is the main issue and this problem is quite serious for cross-polarized SAR with weak sea-surface backscattering roughness. Recently, a few studies were conducted for TC wind retrieval using an empirical algorithm combining the co- and cross-polarized SAR measurements [33,34]; however, this advanced algorithm needs a priori information from external information on wind, such as from the European Centre for Medium-Range Weather Forecasts (ECMWF).

It is recognized that TC wind-retrieval algorithms have two advantages: comparable accuracy and independence of prior information, which is difficult for conventional algorithms. Machine learning refers to the general term for algorithms that identify patterns from data and use them for simulation, classification, and clustering. It is commonly recognized that machine learning could effectively explore the inherent correlations between variables and then make accurate predictions. With a large number of satellite products, machine learning is possibly aiming at research on satellite oceanography [35], which has been applied to the development of SAR wave-retrieval algorithms [36,37]. Since the S-1 mission started in 2016, a campaign for TC observation [38] has been prompted during the annual hurricane season. Utilizing abundant dual-polarized images taken of TCs, SAR wind retrieval using machine learning is anticipated to improve, especially for reducing the discontinuity of the retrieval at the edge of sub-swaths, which can be directly implemented without external data.

The purpose of this work is to study the applicability of three machine learning methods for TC wind speed retrieval from the SAR images without external data. The remaining part of our work is organized as follows. Section 2 gives the descriptions of S-1 images collocated with auxiliary data, including along-track observations from the stepped-frequency microwave radiometer (SFMR) and the product from the soil moisture active passive (SMAP) radiometer. The methodology of the three machine learning algorithms and the fitted results of the training process—using 20 images collocated with SFMR and SMAP data—are introduced in Section 3. The derivation of the TC wind-retrieval algorithm using machine learning algorithms and validation of the wind retrievals against SFMR

observations and SMAP products are exhibited in Section 4. Finally, Section 5 summarizes the conclusions.

## 2. Datasets

During the Satellite Hurricane Observation Campaign (SHOC), a total of 30 S-1 images acquired in IW- and EW-mode were collected in 2016–2022. These images are processed as dual-polarized (VV and VH) for China's and U.S. coastal waters. The TCs' tracks and maximum wind speeds—from the National Hurricane Center (NHC) of the National Oceanic and Atmospheric Administration (NOAA)—overlaid on the geographic coverage of collected images and represented by black rectangles is presented in Figure 1, i.e., (a) at China's coastal waters and (b) the U.S. coastal waters. In particular, SFMR observations from NOAA hurricane aircraft are available for 20 images, which are valuable sources for the development of SAR wind- [39,40] and rain- [41] retrieval algorithms in TCs. As an example, Figure 2a,b illustrate the VV-and VH-polarized NRCS map of an IW image over TC Hermine on 1 September 2016 at 23:45 UTC, in which the track of the SFMR is highlighted by a red line. Similarly, the calibration maps of an EW image over TC Michael on 10 September 2018 at 23:44 UTC are exhibited in Figure 3.

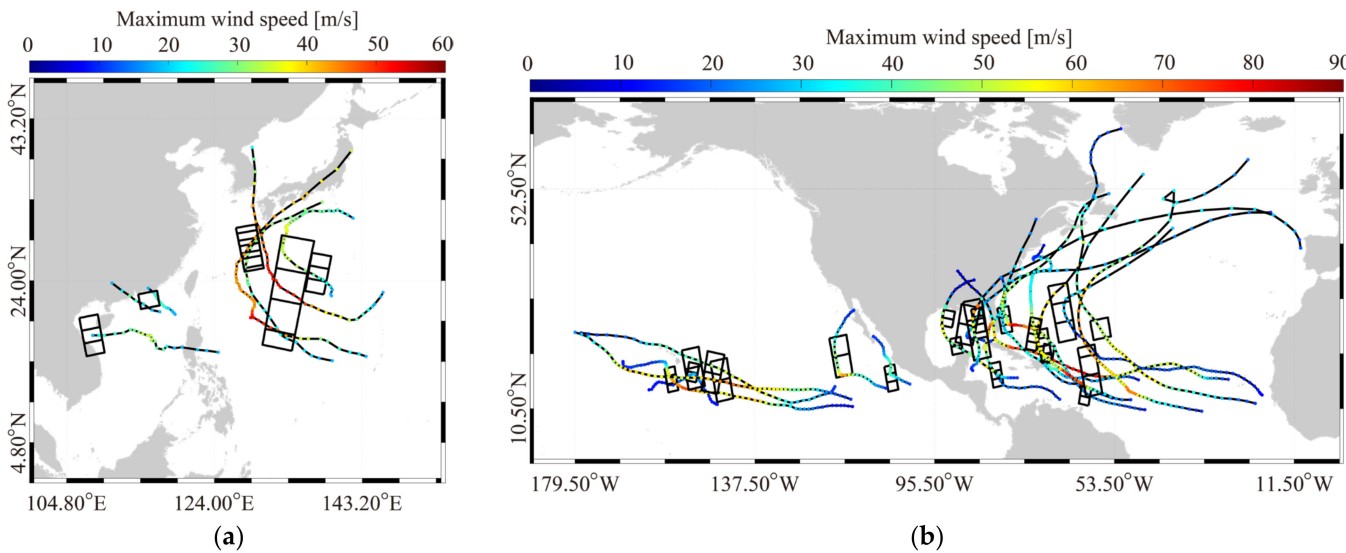

**Figure 1.** Tropical cyclone (TC) tracks and maximum wind speeds from National Hurricane Center (NHC) of National Oceanic and Atmospheric Administration (NOAA) overlaying the geographic coverage of collected Sentinel-1 (S-1) synthetic aperture radar (SAR) images, represented by black rectangles. The images are located at (**a**) China's coastal waters and (**b**) U.S. coastal waters.

The passive L-band radiometer SMAP operating at a sun-synchronous orbit is a National Aeronautics and Space Administration (NASA) mission, active since 2015. Because severe winds can be retrieved from the SMAP-measured brightness temperature, SMAP wind products are beneficial for the analysis of TC intensity and size [42]. Validating SMAP products against SFMR observations for 20 TCs in 2015 and 2016, the standard deviation is ~3 m/s at wind speed greater than 25 m/s [43]. Figure 4a,b show the 0.5° gridded SMAP wind maps over TC Hermine on 1 September 2016 at 23:00 UTC and over TC Michael on 10 September 2018 at 23:00 UTC, respectively.

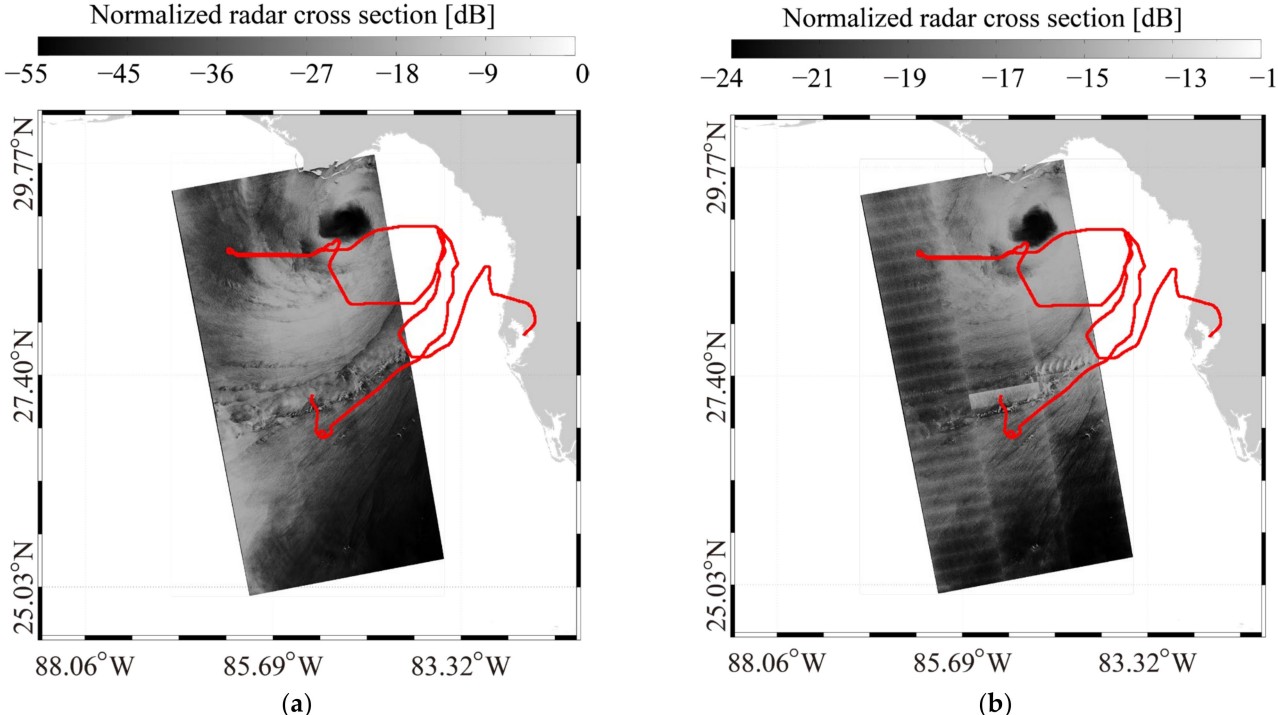

**Figure 2.** The normalized radar cross-section (NRCS) maps of an S-1 interferometric wide (IW) image over TC Hermine on 1 September 2016 at 23:45 UTC: (**a**) vertical–vertical (VV) and (**b**) vertical–horizontal (VH) polarization. The available measurement from stepped-frequency microwave radiometer (SFMR) is highlighted by red line.

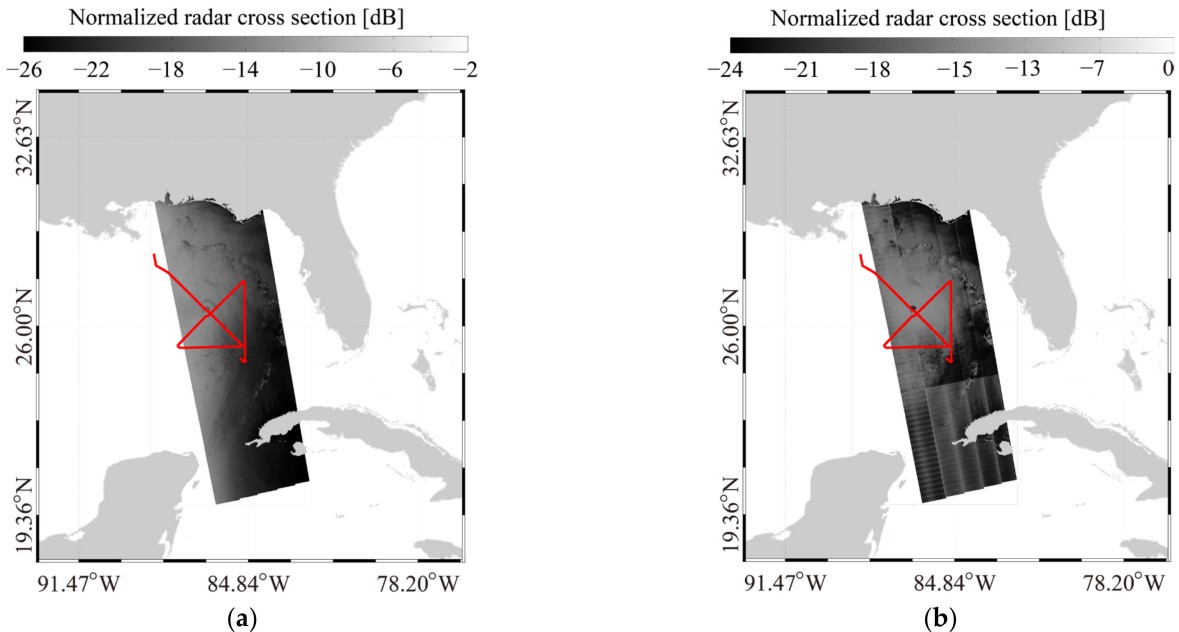

**Figure 3.** The NRCS maps of an S-1 extra-wide (EW) image over TC Michael on 10 September 2018 at 23:44 UTC: (**a**) VV and (**b**) VH polarization. The red lines represent the tracks of SFMR.

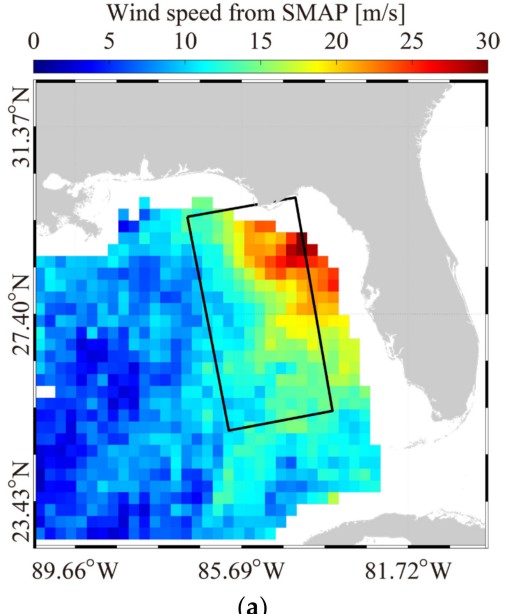

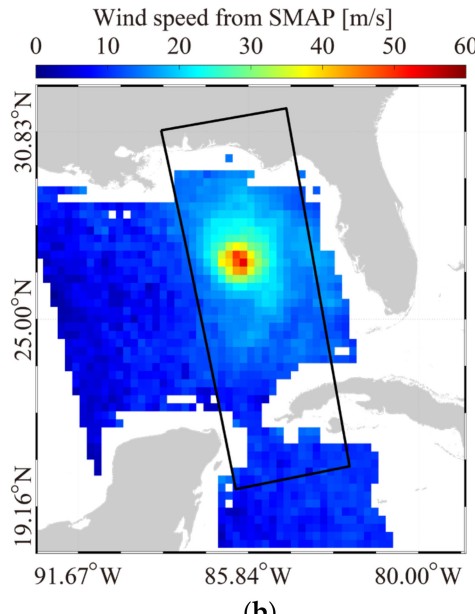

(**a**)                               (**b**)

**Figure 4.** The wind speed maps from soil moisture active passive (SMAP) radiometer over (**a**) TC Hermine on 1 September 2016 at 23:00 UTC, and (**b**) TC Michael on 10 September 2018 at 23:00 UTC. The black rectangles represent the spatial coverages of S-1 IW and EW images in Figures 2 and 3, respectively.

## 3. Machine Learning Algorithms

In this section, three types of machine learning algorithms, i.e., eXtreme Gradient Boosting (XGBoost), Multi-layer Perceptron (MLP), and K-Nearest Neighbor (KNN), are briefly introduced. The performance of the training processes, using two-thirds of the collocated dataset, is also presented.

### 3.1. XGBoost

The eXtreme Gradient Boosting (XGBoost) is an efficient ensemble learning algorithm, which is an open-source gradient lifting framework for machine learning with high efficiency, speed, and precision. The features of XGBoost include a model generated through gradient lifting, the usage of decision trees as weak classifiers, and a final strong classifier through continuous iteration, so as to improve the accuracy of each decision tree. Compared with traditional GBDT, XGBoost utilizes the second-order gradient information and L2 regularization to prevent overfitting and adopts some strategies to speed up the computation.

In principle, XGBoost is a boosting model, that is, each iteration optimizes the submodels in the current step only. It mainly consists of two parts: gradient lifting and regularization. Gradient lifting is an iterative method in which each iterative step adds a weak learner to the model by minimizing the loss function, and regularization is a technique for avoiding over-fitting. The main objective function of XGBoost is expressed:

$$\text{Obj} = \sum_{i=1}^{N} \left[ g_i f_m(x_i) + \frac{1}{2} h_i f_m^2(x_i) \right] + \Omega(f_m) \tag{1}$$

where

$$g_i = \frac{\partial L}{\partial F_{m-1}(x_i)} \tag{2}$$

$$h_i = \frac{\partial^2 L}{\partial^2 F_{m-1}(x_i)} \tag{3}$$

in which $f_m(x_i)$ is the sub-model of the current step; and $F_{m-1}(x_i)$ is the first $m - 1$ sub-model in the training process. The regular term $\Omega(f_m)$ represents the complexity of the sub-model $f$ that is used to control overfitting, and L stands for loss function. As $F_{m-1}(x)$ is determined, $g_i$ and $h_i$ can be easily calculated for each sample point $i$.

To prevent overfitting, XGBoost sets the tree-based complexity as the regular term:

$$\Omega(f) = \gamma T + \frac{1}{2}\lambda\|\omega\|^2 \tag{4}$$

in which $T$ is the number of leaf nodes in tree $f$; $\omega$ is the vector formed by the output regression values of all leaf nodes; $\|\omega\|^2$ is the square of the L2 norm (modulus length) of this vector; and both $\gamma$ and $\lambda$ are hyperparameters. As a regression tree, the more leaf nodes and the larger the output regression value, the higher the complexity of the tree.

Overall, XGBoost is a powerful technique with good scalability, generality, and interpretability. Although it requires some domain knowledge and algorithmic understanding to optimize the model, it is still one of the most popular algorithms in the field of machine learning. Figure 5 shows the scheme of the XGBoost algorithm.

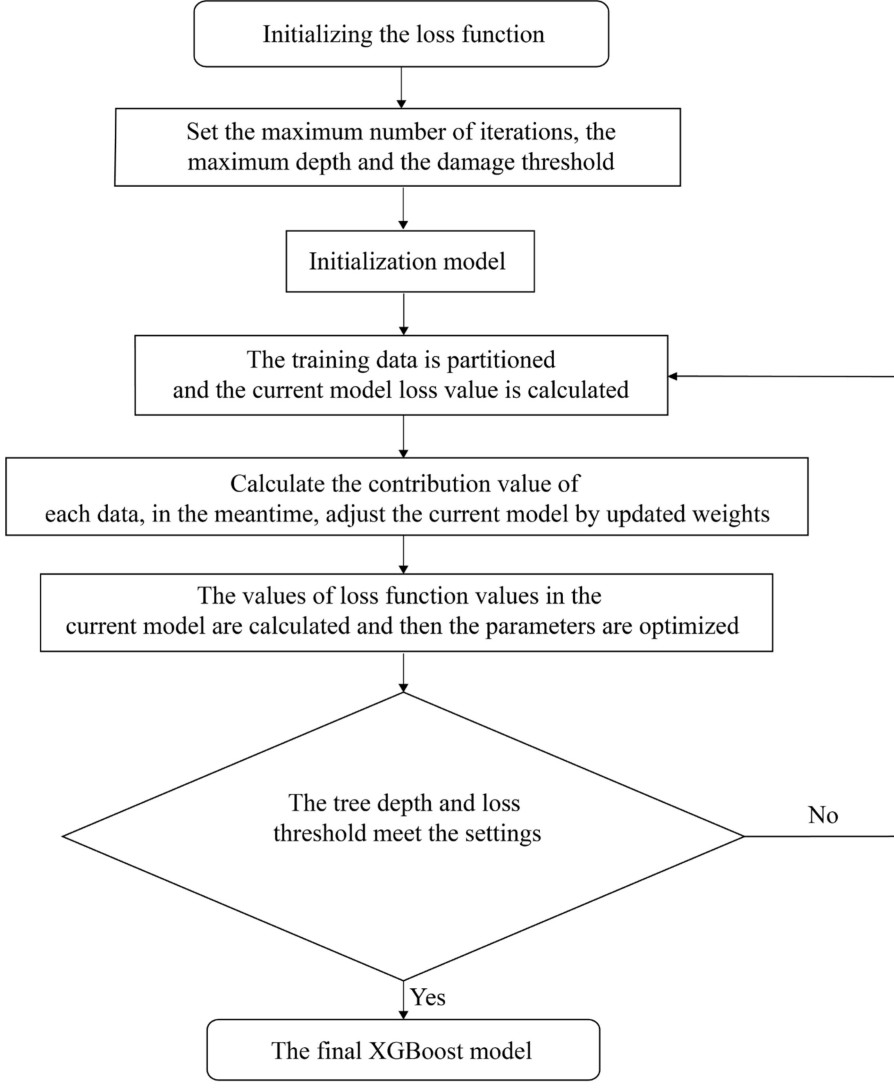

**Figure 5.** The scheme of eXtreme Gradient Boosting (XGBoost).

### 3.2. MLP

MLP is a feedforward neural network model that is widely used in various machine learning and deep learning tasks. In the literature, the MLP model is designed based on the logic structure between neurons, which is an adaptive nonlinear system that can automatically learn the features of input data and establish a mapping relationship between input and output.

In the MLP algorithm, each neural unit receives multiple inputs and a biased term and uses a nonlinear activation function to compute the output. The entire network consists of a series of computation layers containing multiple neurons. The input layer is responsible for receiving the input data and the output layer is responsible for the prediction results of the model. The middle layer, which is also known as the hidden layer, is used to synthesize the features of the input data enabling higher-level feature extraction. The weight parameters of the whole model necessitate it to be optimized through a backpropagation algorithm to minimize the error between the predicted results and the true values of the model.

The basic network structure is shown in Figure 6. The training data are divided into different batches and a set of weights $w$ and bias $\theta$ are defined using these batch cycle training models; the activation function f used in this model is related to the neurons and final outputs, stated as follows:

$$f(z) = \frac{e^z - e^{-z}}{e^z + e^{-z}}, \tag{5}$$

where

$$z = \sum_m w_{mj} + \theta_j, \tag{6}$$

in which $w_{mj}$ is the $m$-th input variable in the neural network and its corresponding $j$-th weight $\theta_j$ is the bias corresponding to the $j$-th neuron.

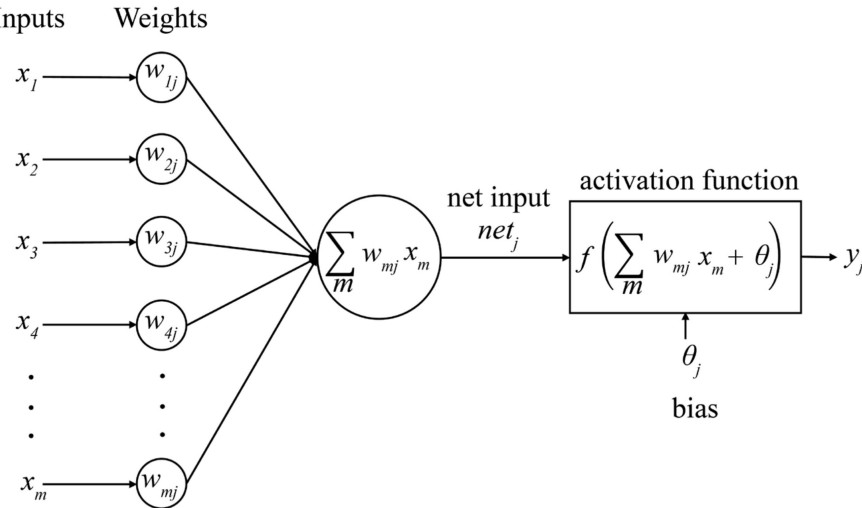

**Figure 6.** The basic network structure of Multi-layer Perceptron (MLP).

### 3.3. KNN

KNN is a parameterless machine learning algorithm, which relies on comparing a new data point to all the data points in the training dataset and then predicting the label or value of the new data point based on the k-most-similar adjacent data points. In fact, KNN is popularly used in classification problems and regression problems. In the regression of the KNN scheme, given a training sample set and a new data point, the algorithm will predict the numerical result of the new data point by calculating the value of the nearest k points around the sample points, which, using the average or weighted averages, predicts the numerical result of the new sample.

KNN mainly consists of the following steps:

1. Determine the number of neighbors, *k*, and the new data points;
2. For a new data point, the distance between it and every data point in the training set is calculated. It is necessary to figure out the distance measurement method used in the KNN algorithm, which depends on the type of data and the relationship between the data. Typically, Euclidean distance is used to measure the distance *d* between two data points. The formulation is described as follows:

$$d = \sqrt{\sum_{i=1}^{m}(x_i - y_i)^2},$$  (7)

3. Select the *k* data point closest to the logarithmic data point. The adjustment of *k* has an important impact on the prediction result of the model. When the value of *k* is small, the model becomes more complex and can fit the training data well but it may overfit; in contrast, if the value of *k* is large, the model becomes simpler;
4. Predict the output of the new data points based on the category (or value) of the nearest neighbor.

The flowchart of the KNN is illustrated in Figure 7.

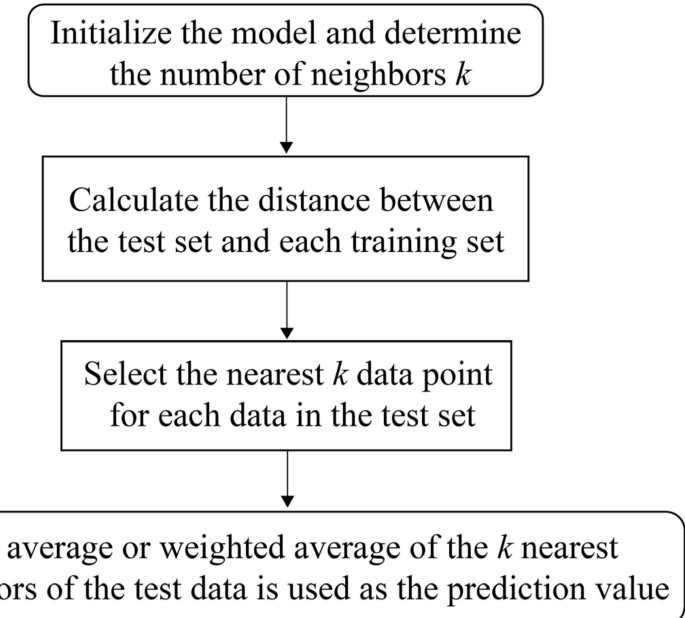

**Figure 7.** The flowchart of K-Nearest Neighbor (KNN).

*3.4. Advanced Wind Dataset*

At present, the French Research Institute for Exploitation of the Oceans (IFREMER) team provides an advanced SAR algorithm to retrieve winds in TCs, which combines the measurements at VV- and VH-polarization channels with a priori information [33,44], such as the ECWMF (its spatial resolution is 0.125° with a time step of 3 h). In this method, the prior information on a wind vector is used and then the retrieval wind is obtained by minimizing the cost function *J*, which is generated using SAR-measured NRCS in VH and VV polarization:

$$J(u,v) = \sum_{\text{pp} \in [\text{VV},\text{VH}]} \left[ \frac{\sigma_0^{\text{pp}} - \text{GMF}^{\text{pp}}(\theta, \phi, U_{10})}{\Delta \sigma_0^{\text{pp}}} \right] + \left[ \frac{u^{\text{priori}} - u}{\Delta u} \right]^2 + \left[ \frac{v^{\text{priori}} - v}{\Delta v} \right]$$  (8)

in which (*u*,*v*) represent the space of solution in the geographical referential from $-80$ to 80 m/s; $u^{\text{priori}}$ and $v^{\text{priori}}$ are the a priori information given by the ECMWF model; $\sigma_0^{\text{pp}}$ and $\Delta \sigma_0^{\text{pp}}$ represent the NRCS measurements in co- (pp = VV) and cross- (pp = VH) polarization and their associated errors; $\text{GMF}^{\text{pp}}(\theta, \phi, U_{10})$ stands for the GMF defined

for each polarization with respect to the incidence angle $\theta$, the wind direction relative to the azimuth direction $\phi$, and wind speed $U_{10}$. The TC wind product denoted as Level-2 CyclObs is also used in our work.

## 4. Results

In this section, the TC wind-retrieval algorithm based on three machine learning algorithms is adopted through 20 images. Then, the validation of retrieved wind speeds from 10 images is exhibited.

### 4.1. TC Wind-Retrieval Algorithm

The key factor for applying machine learning is searching for the variables as the input in the training process. According to backscattering theory, NRCS is determined by wind and this is the foundation of the development of the SAR wind-retrieval algorithm [45]. Under cyclonic conditions, the wave parameters are exponentially related to wind considering the fetch and duration-limited feature inside a TC [46]. on the other hand, the azimuthal cutoff wavelength caused by velocity bunching is determined by the growth of the wave. In this sense, TC wind has a positive relationship with the azimuthal cutoff wavelength [47].

A few sub-scenes extracted from the S-1 image have $256 \times 256$ pixels and $128 \times 128$ pixels with a spatial coverage of 3 km for the EW and IW images, respectively. Because the S-1 IW and EW images are composited by several sub-swathes, the pretreatment for denoise is necessary; therefore, the sub-scenes are smoothed using a $3 \times 3$ Gaussian filter. The azimuthal cutoff wavelength $\lambda_c$ is conveniently calculated by minimizing the standard error of a Gaussian function $G(k_x)$ with respect to wave number $k_x$ in the azimuthal direction, expressed:

$$G(k_x) = \exp\left\{-\pi\left(\frac{k_x}{k_c}\right)^2\right\}, \tag{9}$$

where

$$\lambda_c = \frac{2\pi}{k_c} \tag{10}$$

Note that velocity bunching is independent of polarization; thus, the azimuthal cutoff wavelength in VV-polarization is only used here. In total, three variables are estimated from the sub-scenes, i.e., VV-polarized $\sigma_0^{VV}$ and VH-polarized NRCS $\sigma_0^{VH}$, and the VV-polarized cutoff wavelength $\lambda_c$. There are more than 10,000 samples collocated with wind speed $U_{10}$ from SFMR and SMAP, which is treated as the training dataset. Figure 8 shows the relations between $U_{10}$ and three variables, i.e., (a) $\sigma_0^{VV}$, (b) $\sigma_0^{VH}$, and (c) $\lambda_c$. It is not surprising that the saturation of $\sigma_0^{VV}$ and $\lambda_c$ is observed at high winds, this is because wind speed and SWH are explicitly related, considering the fetch and duration-limited features inside a TC [48]; however, this behavior is not found in relation with respect to $\sigma_0^{VH}$.

In the training procedure, the input parameters include $\sigma_0^{VV}$, $\sigma_0^{VH}$, $\lambda_c$, and incidence angle $\theta$ and the output is $U_{10}$. Figure 9 shows the performance of the training process, i.e., (a) XGBoost and (b) MLP; however, the KNN has no such process. It is observed that the three machine learning algorithms will eventually converge and the RMSE achieved using XGBoost is less than 2 m/s, which is better than the 4 m/s RMSE achieved using MPL.

### 4.2. Validation

The inverted wind maps from the IW image over TC Maria on 21 September 2017 at 22:45 UTC using the three retrieval algorithms are shown in Figure 10, i.e., (a) XGBoost, (b) MLP, and (c) KNN. In the meantime, winds are also retrieved from the VH-polarized image using the GMF S-1 IW Mode Wind Speed Retrieval Model after Noise Removal (S1IW.NR) [40], as presented in Figure 10d. Similarly, the inverted wind maps from the IW image TC Larry 7 September 2021 at 21:48 UTC using XGBoost, MLP, KNN, and the GMF S-1 EW Mode Wind Speed Retrieval Model after Noise Removal (S1EW.NR) [39,49], are illustrated in Figure 11a–d, respectively. It is surprising that the cyclonic structure using KNN

is unclear because there is no iterative regression process; furthermore, the discontinuity of retrievals at the edge of sub-swaths seriously occurs using VH-polarized GMFs.

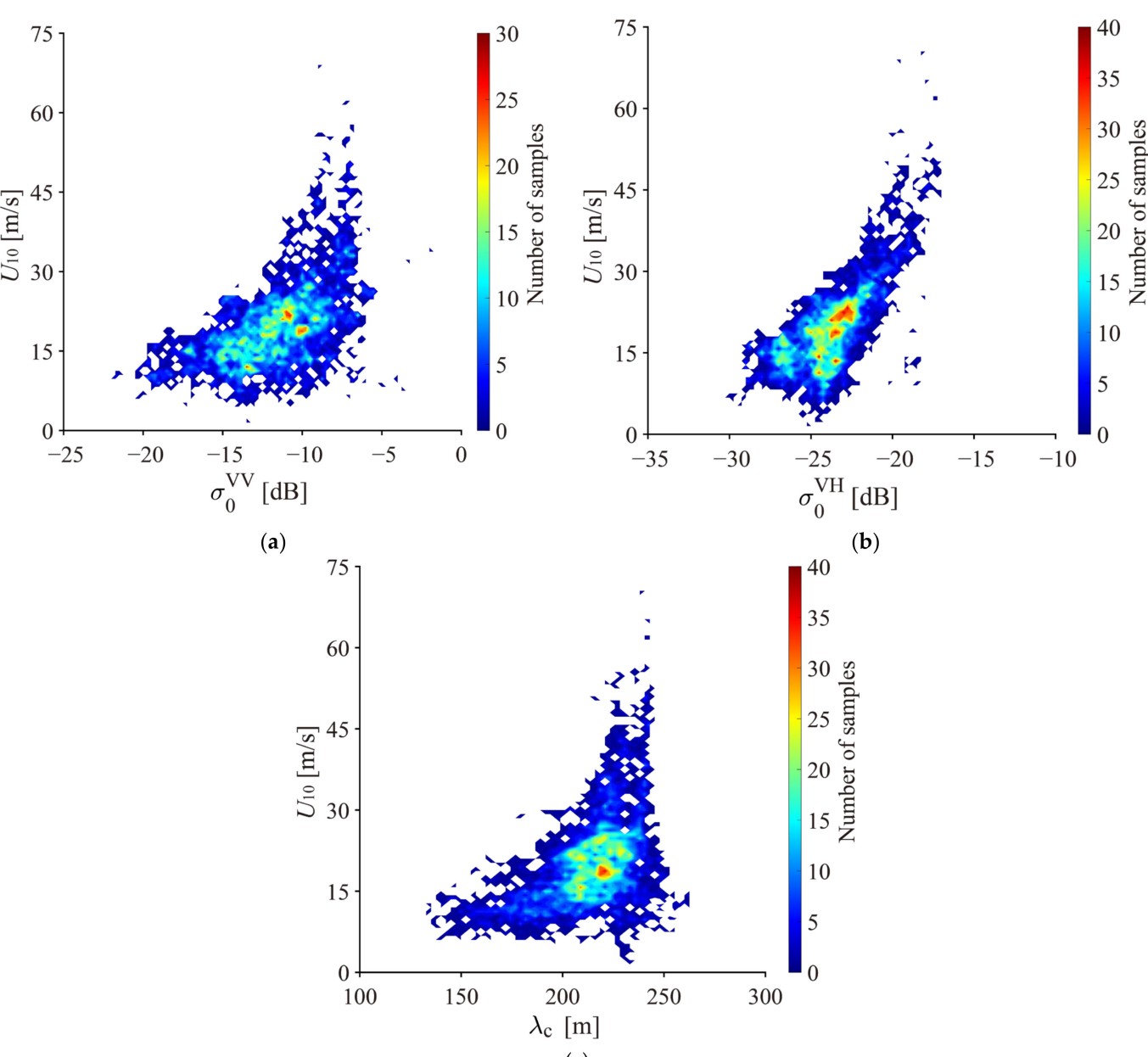

**Figure 8.** Wind speed versus three variables: (**a**) VV-polarized NRCS; (**b**) VH-polarized NRCS; and (**c**) VV-polarized cutoff wavelength.

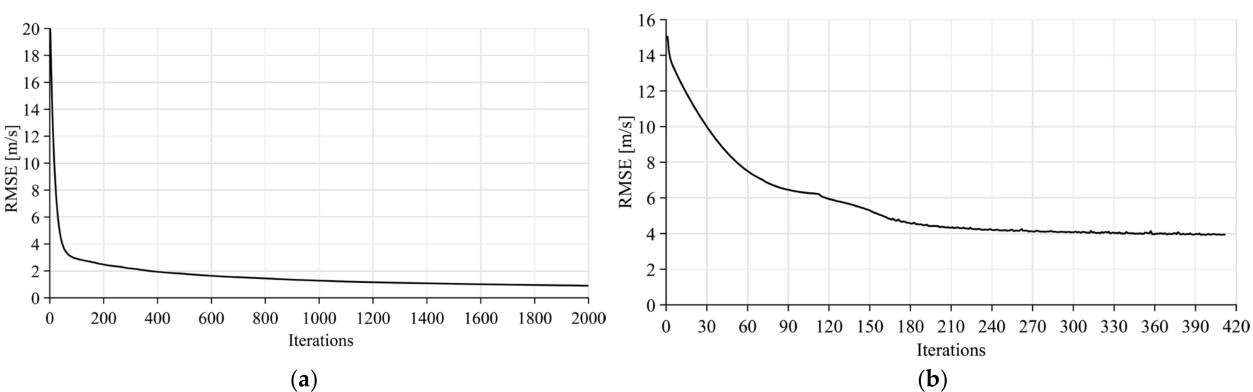

**Figure 9.** The performance of the training process: (**a**) XGBoost and (**b**) MLP.

**Figure 10.** The inverted wind maps from IW images over TC Maria on 21 September 2017 at 22:45 UTC using three machine learning algorithms, i.e., (**a**) XGBoost, (**b**) MLP, (**c**) KNN; and (**d**) the VH-polarized geophysical model function (GMF) denoted as S-1 IW Mode Wind Speed Retrieval Model after Noise Removal (S1IW.NR).

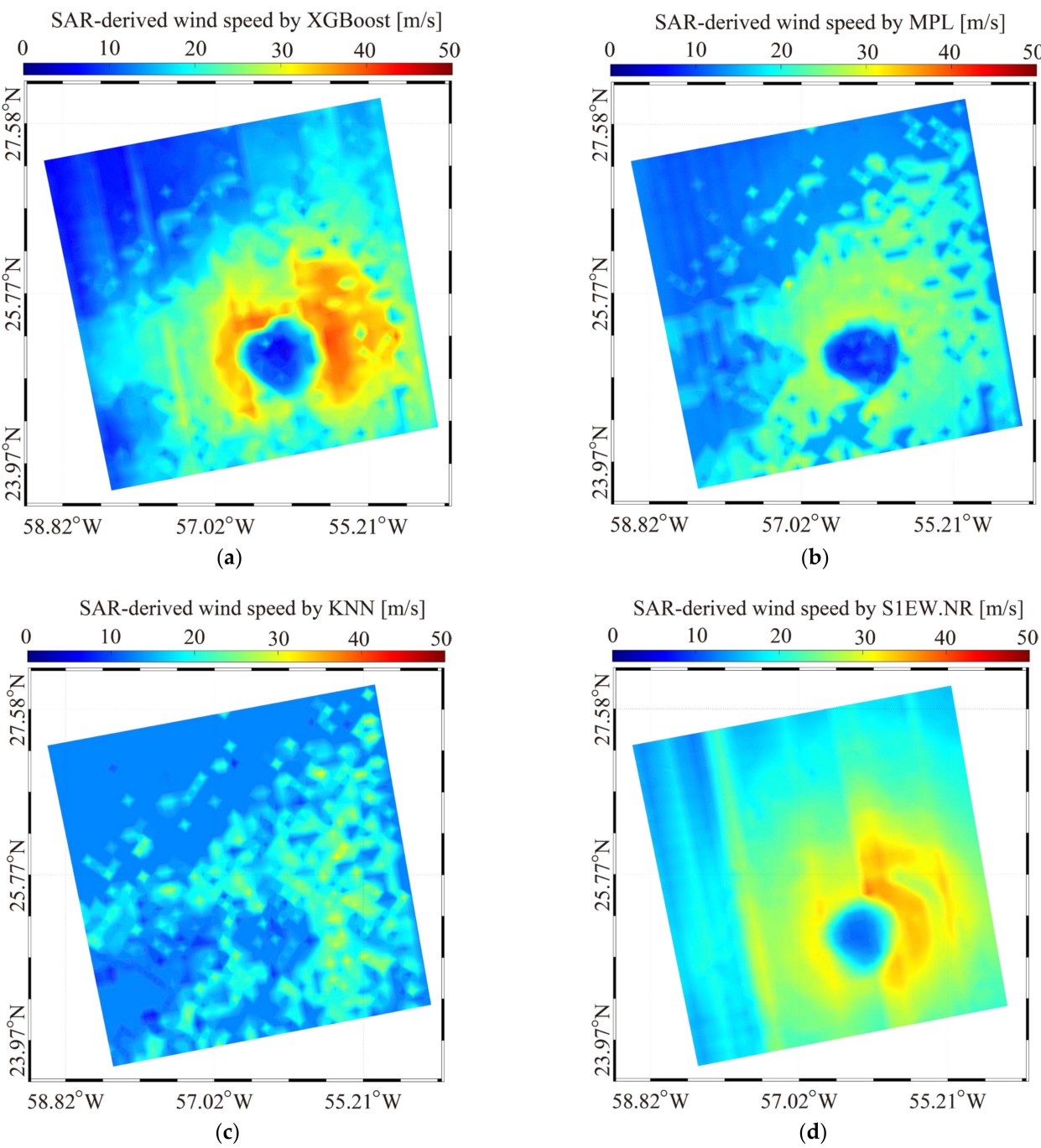

**Figure 11.** The inverted wind maps from EW images over TC Larry 7 September 2021 at 21:48 UTC using three machine learning algorithms, i.e., (**a**) XGBoost, (**b**) MLP, (**c**) KNN; and (**d**) the VH-polarized GMF denoted as S-1 EW Mode Wind Speed Retrieval Model after Noise Removal (S1EW.NR).

Three machine learning algorithms and cross-polarized GMFs are implemented for 10 images. The wind retrievals are validated against the SMAP and SFMR products up to 70 m/s. The following formula is used to calculate the scatter index (SI):

$$\text{SI} = \frac{1}{\overline{X}} \sqrt{\frac{\sum_{i=1}^{n} \left[ \left( X_i - \overline{X} \right) - \left( Y_i - \overline{Y} \right) \right]^2}{n}} \tag{11}$$

It is found that the RMSE of wind speed is 2.53 m/s with a 0.96 correlation (COR) and 0.12 SI using XGBoost (Figure 12a), which is better than the results using other algorithms, i.e., a 5.98 m/s RMSR with a 0.75 COR and a 0.29 SI using MPL (Figure 12b), a 6.41 m/s RMSR with a 0.69 COR and a 0.31 SI using KNN (Figure 12c), and an 8.76 m/s RMSE with a 0.57 COR and a 0.40 SI using cross-polarized GMFs (Figure 12d); therefore, it is believed that XGBoost is the optimal algorithm based on machine learning for inverting TC wind from a dual-polarized S-1 image.

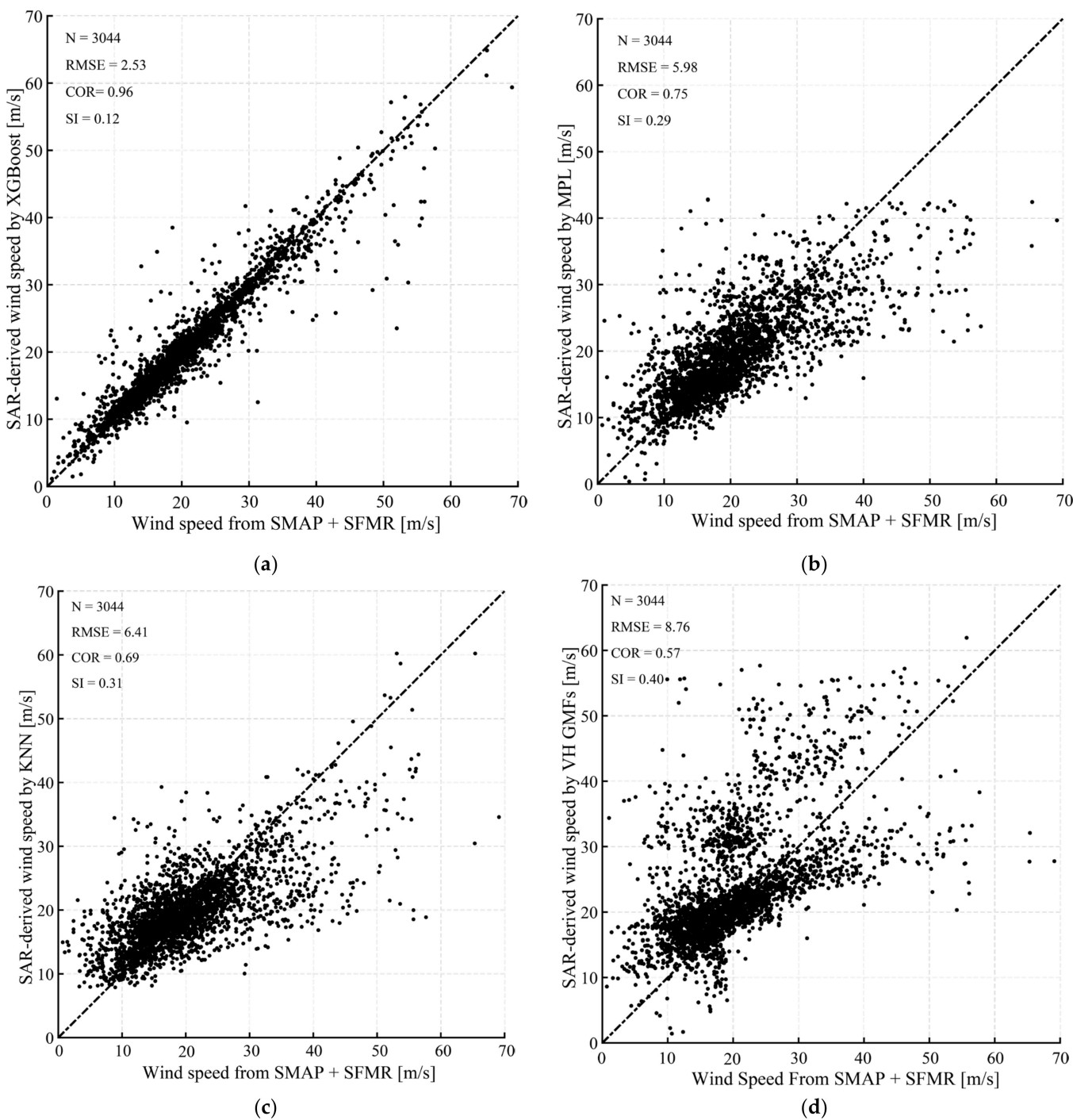

**Figure 12.** Comparisons between the combinations of SMAP and SFMR and SAR-derived wind speeds using four algorithms: (**a**) XGBoost, (**b**) MLP, (**c**) KNN, and (**d**) VH-polarized GMFs.

The advanced CyclObs wind product was also used to evaluate the SAR-derived wind speed using the XGBoost method. The wind map from the CyclObs wind product over TC Maria on 21 September 2017 at 22:45 UTC is shown in Figure 13a, in which the black rectangle represents the spatial coverage seen in Figure 11. It is found that the XGBoost retrieval result is similar to the CyclObs wind product, i.e., cyclonic structure and maximum wind speeds. Moreover, the CyclObs products of 10 images are validated against the collocated matchups from SMAP and SFMR, showing a 4.48 m/s RMSE with a 0.22 SI, as presented in Figure 13b. This is better than the result in Figure 12d because the algorithm generating the CyclObs products is more advanced than previous VH-polarized algorithms. Figure 14 shows the comparison between the CyclObs wind products and the SAR-derived wind speeds using the herein algorithm, in which the wind speed is grouped into a 0.07 m/s bin. The RMSE of wind speed is about 4 m/s with a 0.92 correlation and 0.18 SI. Therefore, we think the XGBoost is the optimal algorithm for inverting TC wind from a dual-polarized S-1 image without external data.

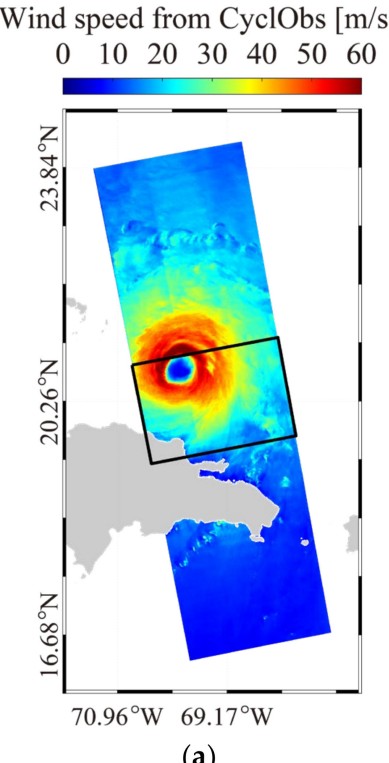

(**a**)

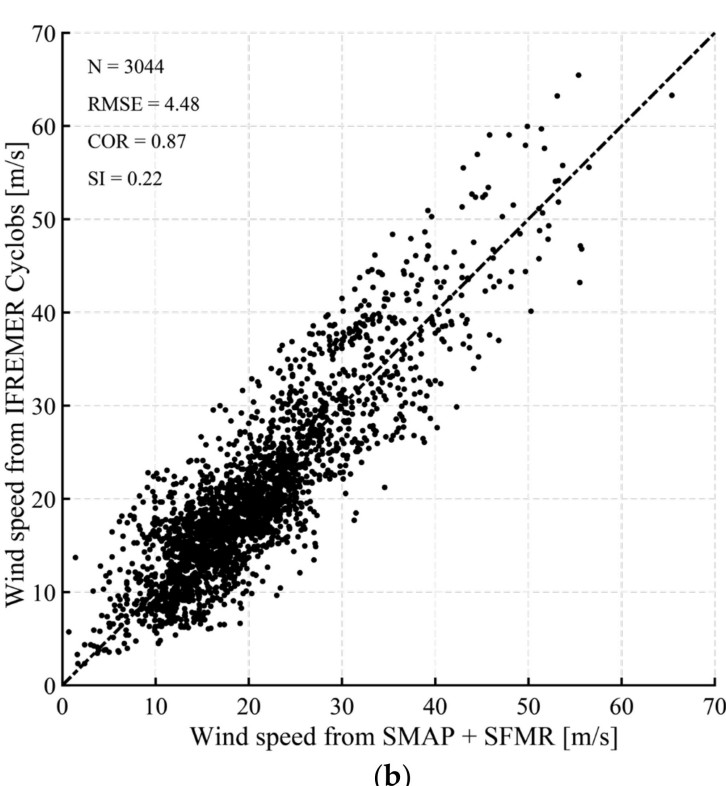

(**b**)

**Figure 13.** (**a**) The wind maps from CyclObs wind product over TC Maria on 21 September 2017 at 22:45 UTC, in which black rectangle represents the position of Figure 11; (**b**) Comparisons between the combination of SMAP and SFMR and CyclObs wind product.

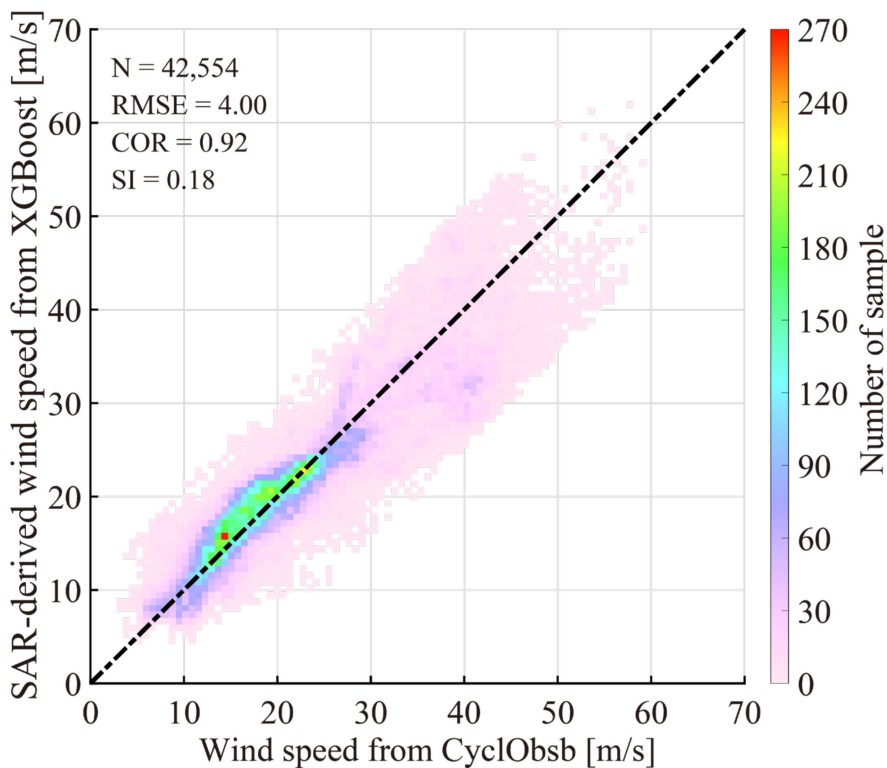

**Figure 14.** Validation of SAR-derived wind speed using XGBoost against the CyclObs wind product, in which the wind speed is grouped into a 0.07 m/s bin.

## 5. Conclusions

Accurate retrieval of TC winds and waves remains a hot topic for the SAR community. Although several GMFs aim to cross-polarize SAR wind retrieval in TCs, discontinuity in the retrieval caused by different NESZs at the edge of sub-swaths is apparently observed. In addition, the accuracy of cross-polarized SAR wind retrieval is relatively lower than that in co-polarization. With a background in artificial intelligence, in this study, the machine learning method has been applied and aimed to solve this problem through abundant data.

In total, 30 S-1 IW/EW images were collected during SHOC in the past 5-year mission of S-1, which were collocated with SMAP products. In particular, SFMR observations aboard NOAA hurricane aircraft are available for 20 images. Theoretically, SAR roughness and the azimuthal cutoff wave length are related to sea-surface wind speed. This is also demonstrated by analyzing the collocated dataset. Three algorithms based on machine learning, i.e., XGBoost, MLP, and KNN were trained through the matchups with $U_{10}$ from SFMR and SMAP, in which $\sigma_0^{VV}$, $\sigma_0^{VH}$, $\lambda_c$, and $\theta$ are set as inputs in order to fit the coefficients. Three machine learning algorithms and VH-polarized GMFs are implemented for the other 10 images collocated with the SFMR observations and SMAP products. It is found that the accuracy of inversion using XGBoost has good performance up to 70 m/s, indicating a 2.98 m/s RMSE of wind speed with a 0.94 COR and 0.14 SI, which is significantly better than the results (> 5 m/s RMSE) from the other two algorithms and the VH-polarized GMFs. It is necessary to figure out if the cyclonic structure of the inverted wind using KNN is unclear, in which case there is no iterative regression process. Again, the discontinuity of retrievals at the edge of sub-swaths is improved using XGBoost, which is inevitably achieved using VH-polarized GMFs. Moreover, the retrieval results from XGBoost have good performance with the CyclObs wind product up to 70 m/s, indicating about 4 m/s RMSE of wind speed with a 0.92 COR and 0.18 SI. Collectively, the XGBoost is the optimal algorithm for inverting TC wind from dual-polarized S-1 images without external data.

In the near future, more SAR images at the C-band will be collected during the hurricane season [49,50], and more tropical cyclonic parameters, i.e., maximum wind speed

and the radius of maximum wind speed will be considered in the training process of machine learning methods.

**Author Contributions:** Conceptualization, W.S. (Weizeng Shao) and Y.H.; methodology, W.S. (Wei Shen) and Y.Z.; validation, Y.H.; formal analysis, W.S. (Weizeng Shao); investigation, W.S. (Weizeng Shao) and W.S. (Wei Shen); resources, W.S. (Weizeng Shao); writing (original draft preparation), W.S. (Wei Shen) and X.J.; writing (review and editing), W.S. (Weizeng Shao) and W.S. (Wei Shen); visualization, Y.H.; funding acquisition, W.S. (Weizeng Shao) and X.J. All authors have read and agreed to the published version of the manuscript.

**Funding:** This research was partly supported by the National Key Research and Development Program of China [2023YFE0102400], the National Natural Science Foundation of China [42076238, 42176012, and 42130402], and the Natural Science Foundation of Shanghai [23ZR1426900].

**Data Availability Statement:** Due to the nature of this study, participants did not agree to their data being shared publicly; therefore, supporting data are unavailable.

**Acknowledgments:** We are thankful for the provision of the Sentinel-1 (S-1) synthetic aperture radar (SAR) images from the European Space Agency (ESA) via https://scihub.copernicus.eu (accessed on 3 August 2023) free of charge. Along-track observations from stepped-frequency microwave radiometers (SFMR) and information on tropical cyclones are released by the National Oceanic and Atmospheric Administration (NOAA). Operational wind production from soil moisture active passive (SMAP) radiometers generated using Remote Sensing Systems were collected via http://www.remss.com (accessed on 3 August 2023). The CyclObs wind production generated by the French Research Institute for Exploitation of the Oceans (IFREMER) team was collected via https://cyclobs.ifremer.fr/app/ (accessed on 3 August 2023).

**Conflicts of Interest:** The authors declare no conflict of interest.

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
