# Peer review of "Machine Learning Applied to a Dual-Polarized Sentinel-1 Image for Wind Retrieval of Tropical Cyclones"

_remotesensing, doi:10.3390/rs15163948_

Round 1
Reviewer 1 Report
This paper developed three machine-learning-based algorithms (XGBoost, MLP, and KNN) for retrieving tropical cyclone winds with SAR observations. An evaluation using SMAP data shows that the XGBoot algorithm has a highest correlation coefficient and smallest root mean square error. I think this paper is publishable but it does need some further work before I could recommend it with enthusiasm.
My concern is the evaluation process. SMAP wind speed retrievals are generally accurate in the range 25–150 kt (12–75 m/s) with errors of 3.5–9 kt (1.8–4.5 m/s). However, the effective extreme wind speeds are limited by sensor resolution and the algorithms' training methods (Meissner et al., 2017). I am not sure whether the evaluation data have enough samples of high and meanwhile accurate wind speeds. Besides, the authors used a SAR wind dataset based on VH-polarized GMFs. I don’t think this dataset is advanced enough to serve as a benchmark. SAR wind data from NOAA (at https://www.star.nesdis.noaa.gov/socd/mecb/sar/AKDEMO_products/APL_winds/tropical/) and IFERMER (at https://cyclobs.ifremer.fr/app/) are recommended.
Meissner, T., Ricciardulli, L., Wentz, F.J., 2017. Capability of the SMAP mission to measure ocean surface winds in storms. Bull. Amer. Meteor. Soc. 98, 1660e1677. https://doi.org/10.1175/BAMS-D-16-0052.1.
Specific comments:
Abstract: The abbreviations (IW, EW, NRCS, MLP, KNN, RMSE, COR, SI, GMF) should be removed since they did not appear again in the abstract section.
Line 25: SAMPàSMAP
Line 83: Section gives à Section 2 gives
Line 123: over (a) over à over (a)
Lines 267–274: The abbreviations (COR and SI) should be given full name when first appeared in the text. Besides, a brief instruction on how to calculate SI is suggested.
Author Response
We thank the reviewers for their constructive comments, which greatly helped us to improve this manuscript. We have provided point-by-point replies herein to carefully address these comments and suggestions.
Reply to Reviewer 1
General comment: This paper developed three machine-learning-based algorithms (XGBoost, MLP, and KNN) for retrieving tropical cyclone winds with SAR observations. An evaluation using SMAP data shows that the XGBoot algorithm has a highest correlation coefficient and smallest root mean square error. I think this paper is publishable but it does need some further work before I could recommend it with enthusiasm.
Reply: We are very grateful to the reviewers for their comments and for the sentences explaining the corresponding problems.
Comment 1: My concern is the evaluation process. SMAP wind speed retrievals are generally accurate in the range 25–150 kt (12–75 m/s) with errors of 3.5–9 kt (1.8–4.5 m/s). However, the effective extreme wind speeds are limited by sensor resolution and the algorithms' training methods (Meissner et al., 2017). I am not sure whether the evaluation data have enough samples of high and meanwhile accurate wind speeds. Besides, the authors used a SAR wind dataset based on VH-polarized GMFs. I don’t think this dataset is advanced enough to serve as a benchmark. SAR wind data from NOAA (at https://www.star.nesdis.noaa.gov/socd/mecb/sar/AKDEMO_products/APL_winds/tropical/) and IFERMER (at https://cyclobs.ifremer.fr/app/) are recommended.
Meissner, T., Ricciardulli, L., Wentz, F.J., 2017. Capability of the SMAP mission to measure ocean surface winds in storms. Bull. Amer. Meteor. Soc. 98, 1660e1677. https://doi.org/10.1175/BAMS-D-16-0052.1.
Reply: For validation dataset, we used the 10 validation SAR images to match the SFMR measurements and got the more than 1000 matchups. We redraw the Figure 12 to show the validation result between the SAR-derived and SMAP + SFMR data. As below:
Page 11: Three machine learning algorithms and cross-polarized GMFs were implemented for 10 images. The wind retrievals were validated against the SMAP and SFMR products up to 70 m/s. The following formula was used to calculate the scatter index (SI):
(11)
It was found that the RMSE of wind speed was 2.53 m/s with a 0.96 correlation (COR) and a 0.12 SI by XGBoost (Figure 12a), which is better than the results by other algorithms, i.e., a 5.98 m/s RMSR with a 0.75 COR and a 0.29 SI by MPL (Figure 12b), a 6.41 m/s RMSR with a 0.69 COR and a 0.31 SI by KNN (Figure 12c), and an 8.76 m/s RMSE with a 0.57 COR and a 0.40 SI by cross-polarized GMFs (Figure 12d). Therefore, it can be concluded that XGBoost is the optimal algorithm based on machine learning for inverting TC wind from dual-polarized S-1 images.
For advance SAR retrieval dataset, we add the section 3.4 and a new Figure 13 to show the comparison result between the SAR-derived wind speed by XGBoost and the CyclObsb wind product from IFERMER (at https://cyclobs.ifremer.fr/app/). It is found that RMSE of wind speed is 4.00 m s-1 with a 0.92 correlation and 0.18 SI.
Page 14: The advanced CyclObsb wind product was also used to evaluate the SAR-derived wind speed by the XGBoost method. The wind map from the CyclObsb wind product over TC Maria on 21 September 2017 at 22:45 UTC is shown in Figure 13a, in which the black rectangle represents the spatial coverage in Figure 11. It was found that the XGBoost retrieval result was similar to the CyclObsb wind product, i.e., cyclonic structure and maximum wind speed. Figure 13b shows the comparison between the CyclObsb wind products and the SAR-derived wind speeds, in which the wind speed is grouped into a 0.07 m/s bin. The RMSE of wind speed was about 4 m/s with a 0.92 correlation and a 0.18 SI. This is relatively worse due to the lack of enough data matching in high wind speed in the SMAP and SFMR datasets. Therefore, we believe that XGBoost is an optimal algorithm for inverting TC wind from dual-polarized S-1 images without external data.
Comment 2: Abstract: The abbreviations (IW, EW, NRCS, MLP, KNN, RMSE, COR, SI, GMF) should be removed since they did not appear again in the abstract section.
Reply: We modified it in the revised manuscript.
Comment 3: Line 25: SAMP à SMAP
Reply: We modified it in the revised manuscript.
Comment 4: Line 83: Section gives à Section 2 gives
Reply: We modified it in the revised manuscript.
Comment 5: Line 123: over (a) over à over (a)
Reply: We modified it in the revised manuscript.
Comment 6: Lines 267–274: The abbreviations (COR and SI) should be given full name when first appeared in the text. Besides, a brief instruction on how to calculate SI is suggested.
Reply: We modified it and add the SI calculating formula in the revised manuscript.
Reviewer 2 Report
The manuscript attempts wind retrieval from Dual-polarized Sentinel-1 Image using Machine Learning techniques. The techniques are worth examination with adequate validations. However, the manuscript needs to look into a few technical and general details for further modifications.
General Comments:
The introduction does not introduce the research problem.
The manuscript needs English correction and grammar check.
Line 35-36: Need reference: Moreover, TC is an important ways of heat exchange in mid to high latitude regions.
Line 72-73: There needs connection between these two paragraphs
Technical comment:
1. The manuscript lacks a description section.
2. The manuscript should explain the need for machine learning.
3. Please explain, how are Machine Learning tools beneficial over the traditional wind computation from Sentinel Images.
4. How will the authors build a unique ML tool considering the uniqueness of tropical cyclone characteristics?
The manuscript needs English correction and grammar check.
Line 83: Do you mean Section 2?
Author Response
We thank the reviewer for their constructive comments, which greatly helped us to improve this manuscript. We have provided point-by-point replies herein to carefully address these comments and suggestions.
Reply to Reviewer 2
General comment: The manuscript attempts wind retrieval from Dual-polarized Sentinel-1 Image using Machine Learning techniques. The techniques are worth examination with adequate validations. However, the manuscript needs to look into a few technical and general details for further modifications.
Reply: We are very grateful to the reviewers for their comments and for the sentences explaining the corresponding problems.
Comment 1: The introduction does not introduce the research problem.
Reply: We add the sentence to explain the research problem in the revised manuscript.
Line 2: The purpose of this work is to study the applicability of three machine learning methods for TC wind speed retrieval from SAR images without external data.
Comment 2: The manuscript needs English correction and grammar check.
Reply: The English editing service provided by MDPI is applied for the manuscript.
Comment 3: Line 35-36: Need reference: Moreover, TC is an important ways of heat exchange in mid to high latitude regions.
Reply: We add the reference to Line 35-36 in the revised manuscript. As below:
- Liu, Y.P.; Tang, D.L.; Evgeny, M. Chlorophyll concentration response to the typhoon wind-pump induced upper ocean processes considering air-sea heat exchange. Remote. Sens. 2019, 11, 1825.
Comment 4: Line 72-73: There needs connection between these two paragraphs.
Reply: We add the sentences to combine these two paragraphs in the revised manuscript.
Page 2: Recently, a few studies have been conducted for TC wind retrieval through the use of an empirical algorithmcombining the co- and cross-polarized SAR measurements [33,34]; however, this advanced algorithm needs a priori information from external information on wind, such as the European Centre for Medium-Range Weather Forecasts (ECMWF).
It is recognized that the TC wind retrieval algorithm has two advantages: comparable accuracy and independence with prior information, which are difficult for conventional algorithms. Machine learning refers to the general term for algorithms that identify patterns from data and use them for simulation, classification, and clustering. It is commonly recognized that machine learning can effectively explore the inherent correlation between variables and then make accurate predictions
Comment 5: The manuscript lacks a description section.
Reply: In this study, we have written the description of our work part in the introduction in the revised manuscript.
Page 2: The remaining part of our work is organized as follows. Section 2 gives the descriptions of S-1 images collocated with auxiliary data, including along-track observations from the stepped-frequency microwave radiometer (SFMR) and the products from the soil moisture active passive (SMAP) radiometer. The methodology of the three machine learning algorithms and the fitted results of the training process through 20 images collocated with SFMR and SMAP data are introduced in Section 3. The derivation of the TC wind retrieval algorithm using machine learningalgorithms and the validation of wind retrievals against SFMR observations and SMAP products is exhibited in Section 4. Finally, Section 5 summarizes the conclusions.
Page 3: 2. Description of datasets
Moreover, the Section 4 is divided into two parts: (1) 4.1 TC wind retrieval algorithm and (2) 4.2Validation.
Comment 6: The manuscript should explain the need for machine learning.
Reply: We add the sentences to show the need for machine learning in the revised manuscript.
Page 2: It is recognized that the TC wind retrieval algorithm has two advantages: comparable accuracy and independence with prior information, which are difficult for conventional algorithms. Machine learning refers to the general term for algorithms that identify patterns from data and use them for simulation, classification, and clustering. It is commonly recognized that machine learning can effectively explore the inherent correlation between variables and then make accurate predictions.
Comment 7: Please explain, how are Machine Learning tools beneficial over the traditional wind computation from Sentinel Images.
Reply: The machine learning method has the advantages: (1) the machine learning can fit the features and patterns of data without the need for predefined complex empirical models and this method can get the TC wind speed without external data; (2) the machine learning method has stronger generalization ability than the traditional wind retrieval method; and (3) the machine learning tools are usually able to obtain more accurate and accurate prediction results through big-data training.
It is recognized that the TC wind retrieval algorithm has two advantages: comparable accuracy and independence with prior information, which are difficult for conventional algorithms. Machine learning refers to the general term for algorithms that identify patterns from data and use them for simulation, classification, and clustering. It is commonly recognized that machine learning can effectively explore the inherent correlation between variables and then make accurate predictions. With a large number of satellite products, machine learning with the aim of conducting research on satellite oceanography is possible [35], and it has been applied to the development of SAR wave retrieval algorithms [36,37]. During the S-1 mission, which started in 2016, a campaign for TC observation [38] was prompted during the annual hurricane season. Utilizing abundant dual-polarized images taken in TCs, it is anticipated that SAR wind retrieval by machine learning will be improved, especially for reducing the discontinuity of the retrieval at the edge of sub-swaths, which can be directly implemented without external data.
Comment 8: How will the authors build a unique ML tool considering the uniqueness of tropical cyclone characteristics?
Reply: In the near future, we will collect more SAR images at C band during hurricane season and use more tropical cyclonic parameters to train the machine learning method. Hereby, the machine learning tool can more adaptable to the TC wind retrieval.
Page 15: In the near future, more SAR images at the C-band will be collected during hurricane season and more tropical cyclonic parameters, i.e., maximum wind speed and the radius of maximum wind speed, will be considered in the training process of machine learning methods.
Round 2
Reviewer 1 Report
I suggest the authors using the SAR wind product by IFREMER CYCLOBS instead of the one by VH-polarized GMFs in Fig. 10d, Fig. 11d and Fig. 12d, because the former is more advanced than the latter. The readers would be more curious to know whether the new algorithm performs better than the algorithm recently used in operation, not an early version.
Fig. 13 just tells the readers how well XGBoost wind product matches with CYCLOBS products, not which one is better.
Specific comments:
CyclObsb --> CyclObs
Page 14, “This is relatively worse due to the lack of enough data matching in high wind speed in the SMAP and SFMR datasets.”:How did the authors get this conclusion?
Author Response
We thank the reviewers for their constructive comments, which greatly helped us to improve this manuscript. We have provided point-by-point replies herein to carefully address these comments and suggestions.
Reply to Reviewer 1
General comment: I suggest the authors using the SAR wind product by IFREMER CYCLOBS instead of the one by VH-polarized GMFs in Fig. 10d, Fig. 11d and Fig. 12d, because the former is more advanced than the latter. The readers would be more curious to know whether the new algorithm performs better than the algorithm recently used in operation, not an early version.
Fig. 13 just tells the readers how well XGBoost wind product matches with CYCLOBS products, not which one is better.
Line 320: The advanced CyclObs wind product was also used to evaluate the SAR-derived wind speed by the XGBoost method. The wind map from CyclObs wind product over TC Maria on 21 September 2017 at 22:45 UTC is shown in Figure 13a, in which black rectangle represent the spatial coverage in Figure 11. It is found that the XGBoost retrieval result is similar to the CyclObs wind product, i.e., cyclonic structure and maximum wind speed. Moreover, the CyclObs products of 10 images are validated against the collocated matchups from SMAP and SFMR, showing a 4.48 m/s RMSE with a 0.22 SI, as presented in Figure 13b. This is better than the result in Figure 12d, because the algorithm generating CyclObs products is more advanced than previous VH-polarized algorithms. Figure 14 shows the comparison between the CyclObs wind products and the SAR-derived wind speeds by herein algorithm, in which the wind speed is grouped into a 0.07 m/s bin. The RMSE of wind speed is about 4 m/s with a 0.92 correlation and 0.18 SI. Therefore, we think the XGBoost is an optimal algorithm for inverting TC wind from dual-polarized S-1 image without the external data.
Reply: We are very grateful to the reviewers for their comments and for the sentences explaining the corresponding problems.
In the revision, we add a new subplot (Figure 13b) in order to compare the CyclObs products and SFMR+SMAP.
Comment 1: CyclObsb --> CyclObs
Reply: We correct it in the revision.
Comment 2: Page 14, “This is relatively worse due to the lack of enough data matching in high wind speed in the SMAP and SFMR datasets.” How did the authors get this conclusion?
Reply: We appreciate this suggestion and we have deleted the ambiguous sentence in the revision.
Round 3
Reviewer 1 Report
It seems that there is a mistake in Figure 13b (same as Figure 14?). Since I can not find a clean version of the revised manuscript, I am not sure whether this is true.
Figure 14: CyclObsb --> CyclObs